# Potential of Molecular Culture in Early Onset Neonatal Sepsis Diagnosis: A Proof of Principle Study

**DOI:** 10.3390/microorganisms11040960

**Published:** 2023-04-07

**Authors:** Thomas Dierikx, Andries Budding, Martine Bos, Henriëtte van Laerhoven, Sophie van der Schoor, Hendrik Niemarkt, Marc Benninga, Anton van Kaam, Douwe Visser, Tim de Meij

**Affiliations:** 1Department of Pediatric Gastroenterology, Amsterdam UMC Location University of Amsterdam, 1105 AZ Amsterdam, The Netherlands; 2Amsterdam Gastroenterology Endocrinology Metabolism, 1105 AZ Amsterdam, The Netherlands; 3Department of Neonatology, Amsterdam UMC Location University of Amsterdam, Meibergdreef 9, 1105 AZ Amsterdam, The Netherlands; 4Amsterdam Reproduction & Development, 1105 AZ Amsterdam, The Netherlands; 5Inbiome, 1098 XG Amsterdam, The Netherlands; 6Department of Pediatrics and Neonatology, OLVG, 1061 AE Amsterdam, The Netherlands; 7Department of Neonatology, Máxima Medisch Centrum, 5504 DB Veldhoven, The Netherlands

**Keywords:** early onset sepsis, neonates, molecular culture, diagnosis, IS-pro

## Abstract

Delay in the time-to-positivity of a peripheral blood culture (PBC), the gold standard for early onset neonatal sepsis (EOS) diagnosis, has resulted in excessive use of antibiotics. In this study, we evaluate the potential of the rapid Molecular Culture (MC) assay for quick EOS diagnosis. In the first part of this study, known positive and spiked blood samples were used to assess the performance of MC. In the in vivo clinical study, the second part of this study, all infants receiving antibiotics for suspicion of EOS were included. At initial EOS suspicion, a blood sample was collected for PBC and MC. MC was able to detect bacteria present in the spiked samples even when the bacterial load was low. In the clinical study, MC was positive in one infant with clinical EOS (*Enterococcus faecalis*) that was not detected by PBC. Additionally, MC was positive in two infants without clinical sepsis (*Streptococcus mitis* and multiple species), referred to as contamination. The other 37 samples were negative both by MC and PBC. MC seems to be able to detect bacteria even when the bacterial load is low. The majority of MC and PBC results were comparable and the risk for contamination and false positive MC results seems to be limited. Since MC can generate results within 4 h following sampling compared with 36–72 h in PBC, MC may have the potential to replace conventional PBC in EOS diagnostics in order to guide clinicians on when to discontinue antibiotic therapy several hours after birth.

## 1. Introduction

Early onset neonatal sepsis (EOS), defined as sepsis within the first 72 h of life, has high morbidity and mortality [1,2]. EOS usually results from transplacental and ascending infections from the maternal genital tract within 72 h after birth. The gold standard for sepsis diagnosis is a conventional peripheral blood culture (PBC), but time to positivity can be up to 36–72 h, and thus is of no value to rule out EOS at the time of initial presentation [3]. Since delay in the treatment of EOS may lead to rapid deterioration or even death, antibiotics are often initiated empirically awaiting PBC results. Roughly 5% of all newborns and over 85% of neonates with a gestational age < 30 weeks are exposed to antibiotics empirically directly after birth [4,5,6], while the incidence of culture-proven EOS is only 0.1–1.2% [7,8,9]. In the vast majority of infants that are started on antibiotics empirically, treatment is thus discontinued after 36–72 h if the PBC remains negative. Besides increasing the risk for multidrug resistant infections, this overexposure to antibiotics early in life leads to aberrations in microbial colonization, increasing the risk for adverse long-term outcomes such as necrotizing enterocolitis, asthma and obesity [10]. Additionally, both infants and their mothers need to be hospitalized, often separated from each other, leading to increased unnecessary hospital costs.

To reduce unnecessary hospitalizations and antibiotic treatment in neonatology intensive care units, it is pivotal that rapid diagnostic tools with a high negative predictive value become available to exclude EOS faster [11]. Molecular techniques that directly detect bacterial DNA might circumvent the delay of PBC by providing rapid results. Currently, the available quality of evidence for the application of molecular techniques in EOS is moderate to low for all studied techniques, such as qPCR and 16S rRNA sequencing, due to the inconsistency and imprecision of results [11]. The disadvantage of qPCR testing includes restrictions based on a limited number of microbial targets based on the selected PCR panel. Drawbacks of unrestricted sequencing techniques include high costs, delays in reporting up to one or more days, lack of standardization and complexity of the procedure [12]. A novel broad-scope molecular technique with capacities to avoid this delay is the Molecular Culture (MC; inbiome, Amsterdam, The Netherlands) assay. MC is an unrestricted PCR-based technique that detects and identifies bacterial DNA via the 16S–23S rRNA gene interspace regions, of which the length signature combined with small sequence polymorphisms is specific for microbial species [13,14]. This unrestricted technique allows for the identification of all bacteria up to the species level and generates results within 4 h. Previous studies in adults comparing MC with results of conventional culturing in samples from abscesses and empyema are very promising, demonstrating that MC detected bacteria in 100% of conventional culture-positive samples. Additionally, MC could detect clinically relevant pathogens that were missed by conventional culture [14]. The sensitivity of conventional PBC for the diagnosis of sepsis in neonates is being questioned [15]. In contrast to conventional PBC, MC may detect bacteria in blood even when the bacterial load is low, is not influenced by previous antibiotic exposure and is able to detect species uncultivatable by PBC [14]. Therefore, it is hypothesized that MC may detect more relevant pathogens in infants with suspicion of EOS compared with a conventional PBC, and the main limitation is expected to be the risk for contamination. However, studies investigating the risk for false positive MC results and its potential as a diagnostic test in blood samples from infants suspected of sepsis are lacking. Therefore, we aimed to assess the ability of MC to detect bacteria in vitro using spiked samples and in clinical samples that were previously shown to be positive. Additionally, we aimed to investigate the risk for false positive results and its potential in cord blood and peripheral blood in a clinical cohort of neonates suspected of sepsis. 

## 2. Methods

### 2.1. Part One: Positive Blood Samples and Spiking Experiments

To test the efficacy of the MC method on bacterial DNA isolated from blood samples, we used samples that were previously collected and processed for the molecular detection of bacteria by a panel of specific qPCRs (MARS study) [16]. In this study, the Polaris method was used to enrich bacterial DNA in 5 mL of blood for improved downstream detection. All methods have been described previously [17]. In short, this method first degrades human cells and DNA, while pathogens remain intact. Secondly, the pathogens are pelleted via centrifugation. Then, pathogens are lysed and DNA can be isolated. A total of 15 samples were selected that had previously been found positive for nine different pathogens with either a high load (Ct < 30) or a low load (Ct > 30). DNA was used in the MC assay (inbiome, Amsterdam, the Netherlands) according to the manufacturer’s instructions. Resulting loads as, expressed in Log_2_ Relative Fluorescence Units (RFU), were compared with Ct values. To investigate the relation between the MC load and the Ct values as found by the qPCR panel, a linear regression analysis was performed between Ct values and Log_2_ transformed Relative Fluorescence Units of the MC. Log_2_ transformation was conducted, as Ct value should also be seen as a log_2_ scale, as it represents measurements of the doubling cycles of PCR.

To test the performance of the Polaris method on small volumes of blood, we spiked blood from a healthy volunteer with three different bacterial species. *Staphylococcus haemolyticus, Escherichia coli* and *Proteus mirabilis*, as representative Gram-positive and Gram-negative bacterial species, were grown overnight on blood agar. From these colonies, a suspension was made in phosphate-buffered saline (PBS) of 0.5 McFarland. These suspensions were diluted tenfold in PBS, after which 10 μL of each dilution was added to 6 mL of blood. The spiked blood was split into six portions of 1 mL. Three of these were pre-processed according to the Polaris protocol, as described previously, after which automated DNA extraction was performed on an EasyMAG machine (BioMerieux, Marcy l’Etoile, France) [17]. Three were directly processed with the EasyMAG machine (see below).

### 2.2. Part Two: Clinical Study in Infants with Suspicion of Early Onset Sepsis

In the second part of this study, we performed a clinical study using samples collected from infants with EOS suspicion. In this prospective observational study, we consecutively included all infants started with antibiotics within the first 72 h of life for suspicion of EOS. Participants were recruited in a level 2 center with two locations (OLVG East and West) between July 2020 and June 2021. Prescription of antibiotics for EOS suspicion was conducted according to the Dutch guideline. In this guideline, maternal risk factors and neonatal risk factors or symptoms of EOS are categorized as red flags or minor criteria (Appendix A) [4]. In the presence of 1 red flag and/or ≥2 minor criteria, it is advised to perform a PBC (BD BACTEC™) and initiate antibiotics empirically for suspicion of EOS. The study protocol was approved by the medical ethical committee of the MEC-U (WO 18.020). All parents gave written informed consent. Infants were not eligible in case of a confirmed congenital infection (toxoplasmosis, rubella, cytomegalovirus, syphilis and herpes).

Discontinuation of empiric antibiotics after 36–72 h was considered in case of a negative PBC and when the clinical condition was reassuring in combination with repeated low C-reactive protein (CRP) concentrations, based on the judgement of the treating physician. Infants with a positive PBC for a microorganism considered as a true pathogen were classified as culture-proven EOS cases. Culture-negative infants who, according to the judgement of the treating physician, continued with antibiotics for ≥5 days and had CRP levels ≥ 10 mg/L were defined as clinical EOS cases. All other participants were classified as uninfected infants. Classification of participants as EOS cases or as uninfected infants was performed blinded from the MC results.

Simultaneously to blood collected for conventional PBC at initial EOS suspicion, 1.0 mL of blood was obtained in an ethylenediaminetetraacetic acid (EDTA) tube from term-born infants. We decided not to collect peripheral blood for MC from preterm infants, as this may increase the risk for iatrogenic anemia due to their low circulating blood volume [18]. If it was prenatally known that the infant would start on antibiotics directly after birth and thus would be eligible for participation, an additional blood sample was collected from the umbilical cord from both term- and preterm-born infants. These samples were collected in a standardized manner after sterilization of the umbilical cord, as previously described [19]. Directly after collection, the blood was stored at −80 °C until further handling.

### 2.3. Sample Handling Processing

All participant samples and half of the spiked samples were pre-processed with the Polaris method, as described previously, after which DNA extraction was performed on an EasyMAG machine (BioMérieux) with the Specific A protocol, as described by the manufacturer [17]. DNA was eluted in 70 μL. The MC analyses were performed according to a previously published protocol by the manufacturer [13]. Identified pathogens by MC were identified and quantified with the online analysis platform Antoni (inbiome). Bacteria found in clinical samples were classified as contamination or as clinically relevant by two independent experts (Tim de Meij, Andries Budding), blinded from the other participant characteristics and PBC results.

### 2.4. Statistical Analysis

Baseline characteristics were presented descriptively for EOS cases and uninfected controls separately. Continuous data were presented as means (standard deviation) or medians (interquartile range) depending on the normality of the distribution. Categorical data were presented as the number (percentage). The results of *MC* were compared with the results of conventional PBC for (clinical) EOS cases and uninfected infants. Statistical analyses were performed in R version 4.0.3.

## 3. Results

### 3.1. Part One: Positive and Spiked Blood Samples

To test the efficacy of the MC method on bacterial DNA isolated from blood samples, a total of 15 samples were selected that had previously been found positive for nine different pathogens. The correct pathogen was detected and identified with MC in 14 of the 15 known positive blood samples by conventional blood culture. The sample in which MC did not detect anything was a sample with a low load of *S. aureus* (Ct 37.6). All comparisons are shown in Table 1. Regression showed a good correlation between Ct values as found by the qPCR panel and MC load, with an R^2^ of 0.78 with an associated *p* value of 2.92 × 10^−5^ (Figure 1). Spiked samples were tested as blood volumes available for diagnostics from infants suspected of EOS, which are typically low (1 mL). As the Polaris method has been designed to enrich microbial DNA in larger volumes of blood, we tested whether this method would have additional value in these small volumes of blood. The test was performed on three replicates on three different bacterial species, *Staphylococcus haemolyticus*, *Proteus mirabilis* and *Escherichia coli*. The spiked blood was split into six portions of 1 mL. Three of these were pre-processed according to the Polaris protocol, as described previously. The other three were directly processed with the EasyMAG. Pre-processing with the Polaris method showed a strong and comparable increase in measured load for all three bacterial species tested (8.4 fold for *S. haemolyticus*, 8.3 fold for *P. mirabilis* and 7.6 fold for *E. coli*), see Figure 2.

### 3.2. Part Two: Clinical Study in Infants with Suspicion of Early Onset Sepsis

A total of 38 eligible participants started on antibiotics for a suspicion of EOS were included. For all participants, a PBC was performed at the initial sepsis evaluation and before the start of antibiotics. None of the participants were classified as culture-proven EOS as all PBCs were negative. A total of 17 infants (44.7%) were classified as clinical EOS cases, and 21 (55.3%) as uninfected infants. Of the 38 included participants, four cord blood samples and thirty-six peripheral samples were collected for MC analysis. From two participants (one clinical EOS case and one uninfected infant), both cord blood and peripheral blood samples were collected. In two participants (both uninfected infants), only cord blood was collected. In summary, we analyzed nineteen peripheral blood samples and three cord blood samples from the uninfected infants. From the clinical EOS cases, we analyzed seventeen peripheral blood samples and one cord blood sample. Baseline characteristics are given in Table 2.

None of the infants were exposed to antibiotics before the collection of samples for both PBC and MC. PBCs were negative in all 38 participants. MC was positive in three of forty (7.5%) samples. All three positive samples were peripheral neonatal samples. In one infant classified as clinical EOS, *Enterococcus faecalis* was identified by MC. In one participant classified as an uninfected infant, *Streptococcus mitis* was detected, and in another uninfected infant, MC showed multiple species (*Sneathia vaginalis*, *Prevotella bivia*, *Phocaeicola dorei* and *Bacteroides fragilis*). No umbilical cord blood samples were collected from these three participants. In the other 37 of 40 (92.5%) MC samples, results were negative and thus comparable to PBC results. The MC was negative in all four cord blood samples.

## 4. Discussion

In this prospective cohort study, we demonstrated the feasibility of MC to detect bacteria in blood. We demonstrated that pre-processing with the Polaris method showed improved detection of bacteria, even in low volumes of blood. Furthermore, we investigated the potential of MC as diagnostic test for EOS. Bacteria were detectable by MC in spiked and known positive samples, even when present in low concentrations. All conventional PBCs of included infants were negative and MC results were similar in 92.5% of samples. MC detected *Enterococcus faecalis* in one clinical EOS case that was missed by PBC, and was positive in two uninfected infants, which are suspected to be false positives.

Diagnostic tools with rapid turnaround time and a high negative predictive value are needed to safely decrease excessive use of antibiotics in unaffected infants suspected of EOS. In the past decades, molecular-based diagnostic tests have become available for identification of bacterial DNA, such as real time PCR, 16S rRNA gene sequencing and MC [11,20]. qPCR techniques are restricted by the chosen panel, so they only detect a pre-defined set of bacteria [21]. Unrestricted techniques such as 16S sequencing are expensive and have a reporting delay of one to several days [12]. MC, on the other hand, is an unrestricted technique that allows for identification of all bacteria to the species level and generates results within 4 h. In contrast to conventional PBC, this molecular technique is not influenced by maternal intrapartum antibiotic prophylaxis and is able to detect species not detectable by PBC [14]. Consequently, the sensitivity of MC for EOS might be higher compared with a conventional PBC. On the other hand, this sensitive method also increases the risk for false positive results. Here, we demonstrated that MC is able to detect bacteria present in low loads using spiked and known positive samples.

In our study including infants with EOS suspicion, all conventional PBC results were negative. MC results were comparable to the PBC in the majority of blood samples. Additionally, MC allowed for the detection of *Enterococcus faecalis* in one clinical EOS case that had negative PBC. Notably, *Enterococcus faecalis* is a microorganism which is difficult to detect using conventional techniques [22,23], illustrating the limited sensitivity of standard PBC. The risk for false positive MC results in the peripheral blood and cord blood of neonates suspected of EOS seemed to be limited, as only two other samples were positive by MC.

Discrepancies between MC results and PBC results can be explained by a number of factors. First, MC can detect certain types of bacteria, both true pathogens and contaminants, that are unable to grow in PBC medium due to peculiar growth requirements [23,24]. This is shown by the positive samples of one clinical EOS case (*Enterococcus faecalis*) and two positive samples from uninfected infants in this cohort. Based on the detected bacteria in uninfected controls (*Streptococcus mitis* and a sample with multiple species associated with vaginal and rectal microbiota), however, these bacteria have been considered to be contaminants. Additionally, PBC results may be false negative in cases of low bacterial loads and previous antibiotic exposure. Furthermore, both tests are at increased risk for false negative results in cases of limited and inadequate sampled blood volume, consequently leading to discrepant results.

New diagnostic tests can either replace the original test, be applied as triage test before the current test or applied as an add-on test to the existing standard [25]. Based on the fast turnaround time of the MC and the potential ability to predict negative PBC results, it might be suitable to added to current EOS guidelines to exclude EOS faster. This could guide clinicians to discontinue antibiotics in cases of negative MC if clinical conditions and other laboratory measures are reassuring within 4 h, instead of after 36–72 h when using PBC. This would decrease the duration of unnecessary antibiotic exposure, reduce unnecessary hospitalization and costs and lead to improvement of microbiota-related short- and long-term outcomes. As there were no positive PBCs in our cohort, we were unable to investigate whether the MC will detect all cultured bacteria by PBC in infants, as demonstrated in a previous study in adults. Here, we demonstrated that the risk for false positive MC results seems to be limited. Before clinical application, the value of MC needs to be validated in larger cohorts including culture-positive EOS cases. However, it has to be acknowledged that Molecular Culture does not provide antibiotic resistance patterns. Therefore, conventional blood cultures will be necessary for this purpose as long as there are no opportunities to establish antibiotic resistance patterns by novel molecular techniques.

Collecting blood for a PBC in infants can be challenging and is a painful procedure. A limited volume is often sampled due to the risk of iatrogenic anemia in infants [26,27,28], but this may increase the risk for false negative results. Collection of blood from the umbilical cord allows sampling of a larger volume, which increases the sensitivity of a blood culture [29]. Previous standard operating manuals have been designed for sterile collection of cord blood [19]. The four cord blood samples collected in this study all had negative MC results. Due to the limited number of cord blood samples, future research is needed to validate that cord blood is of added value for molecular bacterial culturing in EOS diagnostics.

Strengths of this study include the pre-clinical testing of the efficacy of the MC to detect bacteria in known positive and spiked blood samples and the evaluation of the added value of a preprocessing technique that specifically enriches bacterial DNA. The prospective, consecutive inclusions of patients allows for the generalization of results to clinical practice. Furthermore, results of the MC were interpreted blindly from PBC results and other participants’ data. Limitations of this study include the lack of culture-positive EOS cases, hampering the opportunity to investigate whether MC can also predict a positive PBC. Furthermore, clinicians were trained to collect sterile samples, but samples might still have been contaminated during collection or during the analysis. Finally, the sample size of this cohort was relatively small, and limited blood volume availability in infants might have impacted the results of the MC. To further investigate the potential of MC for EOS diagnosis, we are planning to perform a larger study in neonates suspected for sepsis. We also aim to include samples from older infants, children and adults to determine whether this technique may be suitable for sepsis diagnosis in other populations.

## 5. Conclusions

MC was able to detect bacteria even when bacterial load was low in known positive and spiked samples. This is the first study that investigates the risk for false positive MC results and the potential of MC as a diagnostic test in neonates suspected of sepsis. All PBC results and the majority of MC results were negative, so the risk for false positive MC results seems to be limited. MC allowed for the detection of *Enterococcus faecalis* in one clinical EOS case that was missed by PBC, and two positive tests in uninfected infants, considered to be contamination. Since MC can exclude the presence of bacteria in a sample within 4 h following sampling, compared with 36–72 h in PBC, MC may be added to current EOS guidelines. This may guide clinicians to discontinue antibiotic therapy faster in cases of a negative MC test and reassuring clinical condition of the infant. As long as there are no opportunities to establish antibiotic resistance patterns by novel molecular techniques, blood cultures will be necessary for this purpose.

Future prospective studies are needed in larger cohorts containing culture-positive EOS cases to evaluate the accuracy of the rapid MC technique for EOS diagnosis, avoiding the delay characterizing PBC. This could dramatically reduce excessive use of antibiotics in the neonatal population.

## Figures and Tables

**Figure 1 microorganisms-11-00960-f001:**
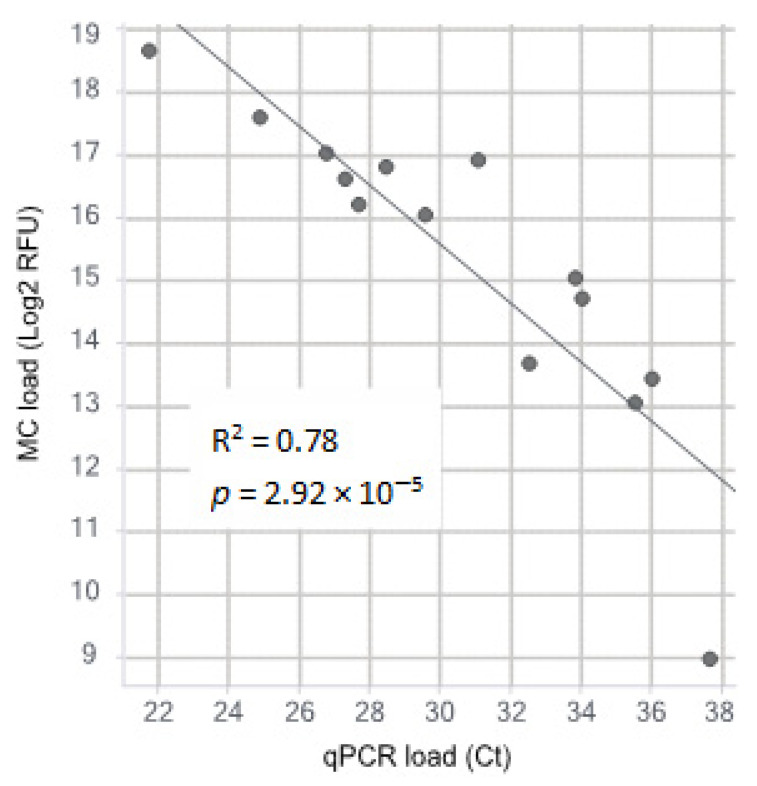
A linear correlation can be seen between Ct values as measured by specific qPCR and MC load, as measured by Log_2_ transformed Relative Fluorescence Units (RFU) (R^2^ = 0.78, *p* = 2.92 × 10^−5^).

**Figure 2 microorganisms-11-00960-f002:**
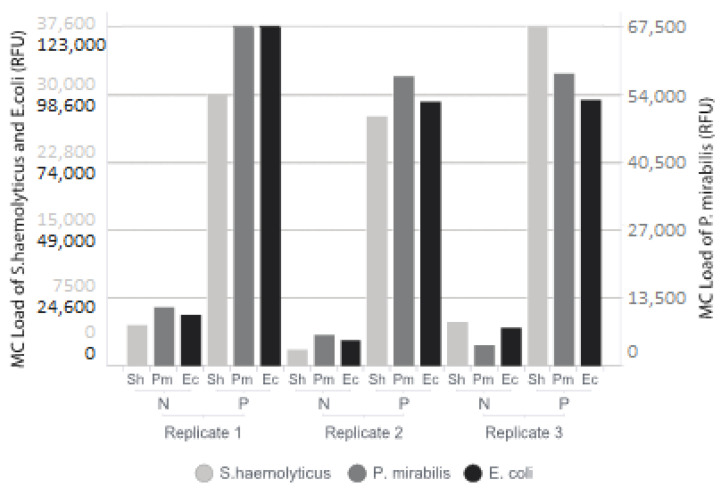
Comparison of DNA extraction with (P) or without (N) Polaris pre-treatment. The test was performed on three replicates on three different bacterial species, *Staphylococcus haemolyticus* (Sh), *Proteus mirabilis* (Pm) and *Escherichia coli* (Ec). Molecular Culture loads are expressed in Relative Fluorescence Units (RFU). Adding Polaris pre-treatment resulted in significantly increased detected loads.

**Table 1 microorganisms-11-00960-t001:** Detection of different bacterial species via molecular culture (MC) with either a high (Ct < 30) or a low (Ct > 30) load isolated from blood with Polaris pre-treatment. MC loads are expressed in Log_2_ Relative Fluorescence Units.

	High		Low	
	Ct	MC Load	Ct	MC Load
*Enterococcus faecalis*	24.89	199,345	35.55	8548
*Enterococcus faecium*	28.46	114,870	36.05	11,143
*Escherichia coli*	26.77	134,370	34.03	27,040
*Klebsiella pneumoniae*	21.75	413,780	33.84	33,867
*Morganella morganii*			32.55	13,174
*Pseudomonas aeruginosa*	29.57	67,570	37.68	500
*Salmonella enteritidis*	27.33	100,043		
*Staphylococcus aureus*	31.1	123,076	37.64	0
*Streptococcus pneumoniae*	27.71	75,570		

**Table 2 microorganisms-11-00960-t002:** Baseline characteristics.

	Controls (n = 21)	Clinical EOS Cases (n = 17)
Gestational age, median [IQR], weeks + days	38^+1^ [36^+0^–40^+6^]	40^+2^ [38^+6^–41^+1^]
Birthweight, median [IQR], grams	3300 [2697–3835]	3676 [3353–4126]
Female gender, n (%)	9 (43%)	4 (24%)
Vaginal delivery, n (%)	6 (29%)	8 (47%)
C-reactive protein, median [IQR], mg/L	6.8 [1.7–17.0]	48.0 [32.5–70.0]
Maternal age, mean (sd), years	32.0 [30.0–34.0]	34.0 [29.3–35.5]
5 min Apgar score, median [IQR]	10 [10–10]	9 [7–10]
Maternal fever, n (%) *	9 (43%)	10 (59%)
Maternal GBS colonization, n (%)	6 (29%)	1 (6%)
PROM, n (%) **	14 (67%)	8 (47%)
Maternal IAP, n (%)	10 (48%)	7 (41%)
Well-appearing at inclusion, n (%) ***	11 (52%)	1 (6%)

* Maternal fever defined as intrapartum temperature >38 °C. ** PROM defined as rupture of membranes >18 h before labor onset after a pregnancy of <37 weeks and >24 h after a pregnancy of ≥37 weeks. *** Asymptomatic infants without (non-specific) clinical signs such as tachypnea, dyspnea and temperature instability started on antibiotics solely based on maternal risk factors for early onset neonatal sepsis. GBS: Group B Streptococcus; IAP: intrapartum antibiotic prophylaxis; IQR: interquartile range; PROM: premature rupture of membranes; sd: standard deviation.

## Data Availability

De-identified data for this study are available from the corresponding author upon reasonable request.

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
