# Peer review of "Potential of Molecular Culture in Early Onset Neonatal Sepsis Diagnosis: A Proof of Principle Study"

_microorganisms, 2023, doi:10.3390/microorganisms11040960_

Round 1

Reviewer 1 Report

Dear authors,

 I recomend to include in the section 2.4. Statistical analysis the description of linear regression analysis cited in lines 156 to 161.

Author Response

Reviewer 1

Dear authors,

I recomend to include in the section 2.4. Statistical analysis the description of linear regression analysis cited in lines 156 to 161.

Dear Reviewer,

Thank you very much for reading of and commenting on our manuscript entitled “Potential of Molecular Culture in Early-onset Neonatal Sepsis Diagnosis: a Proof of Principle Study”. We have addressed the point risen and included the statistical description of the linear regression in the methods section (line 97-101 in the tracked change version of the adapted manuscript).

Reviewer 2 Report

The aspect of as fast as possible diagnostics of sepsis is absolutely pivotal for patients’ lives. So any tests of a new method that would shorten this process are really important. In this context, the studies described are very valuable from both scientific and practical point of view. However, I have some comments, which need to be addressed before the acceptance of the manuscript to be published in Microorganisms.

Major comments:

1. First let me express some ethical objections because of the study groups: the newborns, including, if I well understood, preterm born (the group suspected of systemic infection with possibility of sepsis development). I saw the number of the medical ethical committee of the MEC-U (WO 18.020) in Materials and Methods, but for the readers it means nothing. So, because of this such ethically ambiguous situation, the copy of ethics committee permission should be included to the text (maybe as Supplementary Materials).

2.    Nomenclature (Abstract and whole text): “a peripheral bacterial culture (PBC)”. As you know the word “peripheral” is referred to blood, so the name of method described as the gold standard is peripheral blood culture. Especially that many different pathogens (not only bacteria) can be expected in blood samples despite the same symptoms.

3.   Abstract and Introduction: PBC is the gold standard not only for EOS, but general for sepsis diagnostics. Improve the sentences

4.     The number of samples (control and tested) is unclear. In Abstract you gave such information: “in one infants … in two infants …The other 37 samples” (giving together 40 samples). In Materials and Methods (part one): “A total of 15 samples were selected”. In Table 2: “Control n=21” “Tested n=17”. Moreover, you collect one or both: cord or peripheral blood samples. Give the appropriate numbers of control and tested samples in each part of experiments in Materials and Methods (perhaps a diagram would be helpful to the reader).

5.    Abstract/ Introduction/ Discussion/ Conclusion: “Since MC can generate results within 4 hours following sampling compared to 36-72 hours in PBC”; “time to positivity is commonly up to 36-72 hours”; “compared to 36-72 hours in PBC” – The sentences seem to be biased in favor of MC test. Classic microbial culture on solid or liquid media takes 24 hours. If you have blood samples and there is a suspicion of systemic infection with possibility of sepsis the automated blood culture systems are used (like VITEK, BACTEC Alert), which allow to shorten a culture to a few hours. You have to give correct information about blood culture systems and time needed for the culture results.

6.    Materials and Methods: “classified as contamination or as clinically relevant by two independent experts (TdM, DB) – Explain the abbreviations, what kind of experts you are talking about?

7.   The sentence is incomprehensible: “The sample in which MC did not detect anything was a sample with a low load of S. aureus (Ct 37,6)” – Since MC is also based on PCR, you need clearly indicate the methods used (How do you know about S. aureus?).

8.   Discussion: “molecular techniques have become available for identification of bacterial DNA, such as real time PCR, 16S rRNA gene sequencing and MC” – MC is a name of company's diagnostic test (in addition disastrous and completely misleading in my opinion, but nothing can be done about it anymore), not a molecular technique (the test is based on PCR). Improve the sentence.

Minor comments:

1. Whole text: Proper spelling with hyphen: Gram-positive, Gram-negative bacteria.

2. Table 1/ Figures / whole text: The Latin names should be written italics.

3. Whole text: The spaces are missing after dot in short Latin names, e.g. E. coli instead of E.coli

4.  Abstract: What do you mean writing “spiked blood samples” – combined, pooled, mixed?

5. Abstract: “bacteria present in low concentrations” – Bacteria form a suspension, not a solution, so you can't talk about their “concentration”

6. Whole text: “Log2” or Log2 (base of the logarithm)?

7. Whole text: “5ml” “1ml” etc. – lack of space before the units (it should be 5 ml, 1 ml etc.)

8. Materials and Methods: “PBS” -  for the first time give also the full name and the source of buffer

9. Materials and methods: “To test the performance of the Polaris method on small volumes of blood, we spiked 1ml aliquots of blood” – according to your later description you mixed bacterial suspension with 6 ml of blood sample and finally divided and took 1 ml for testing.

10.  Materials and methods / Discussion: “micro-organisms” – redundant dash

11.  inbiome ver. inBiome – unify in whole text

12.  Whole text: Table 1 or 2, Figure 1 or 2 start from the capital letters

13. The legend to the Table 1 is needed: signs/descriptions (e.g. MC load) need to be explained. Clarify in what units did you present MC load”?

Author Response

Reviewer 2

The aspect of as fast as possible diagnostics of sepsis is absolutely pivotal for patients’ lives. So any tests of a new method that would shorten this process are really important. In this context, the studies described are very valuable from both scientific and practical point of view. However, I have some comments, which need to be addressed before the acceptance of the manuscript to be published in Microorganisms.

Dear Reviewer,

Thank you very much for your valuable comments on our manuscript entitled “Potential of Molecular Culture in Early-onset Neonatal Sepsis Diagnosis: a Proof of Principle Study”. Each of your insights have served to strengthen our manuscript and we have made changes to reflect them. Below we present the detailed replies to your comments. Please find also the revised manuscript.

Major comments:

  1. First let me express some ethical objections because of the study groups: the newborns, including, if I well understood, preterm born (the group suspected of systemic infection with possibility of sepsis development). I saw the number of the medical ethical committee of the MEC-U (WO 18.020) in Materials and Methods, but for the readers it means nothing. So, because of this such ethically ambiguous situation, the copy of ethics committee permission should be included to the text (maybe as Supplementary Materials).

Answer/Response of the Authors: The permission letter from the medical ethical committee has been added as supplementary material to the revised manuscript. However, the permission letter is in Dutch. We would like to leave the decision to decide whether this addition is useful to the editor.

  1. Nomenclature (Abstract and whole text): “a peripheral bacterial culture (PBC)”. As you know the word “peripheral” is referred to blood, so the name of method described as the gold standard is peripheral blood culture. Especially that many different pathogens (not only bacteria) can be expected in blood samples despite the same symptoms.

Answer/Response of the Authors: ‘peripheral bacterial culture’ was changed to ‘peripheral blood culture’ (line 17 in the revised manuscript).

  1. Abstract and Introduction: PBC is the gold standard not only for EOS, but general for sepsis diagnostics. Improve the sentences

Answer/Response of the Authors: we adjusted this sentence in the introduction. It is now stated that a peripheral blood culture is the gold standard for sepsis in general (line 39-40 in the revised manuscript).

  1. The number of samples (control and tested) is unclear. In Abstract you gave such information: “in one infants … in two infants …The other 37 samples” (giving together 40 samples). In Materials and Methods (part one): “A total of 15 samples were selected”. In Table 2: “Control n=21” “Tested n=17”. Moreover, you collect one or both: cord or peripheral blood samples. Give the appropriate numbers of control and tested samples in each part of experiments in Materials and Methods (perhaps a diagram would be helpful to the reader).

Answer/Response of the Authors: This study consisted of two parts: part I include the two laboratory experiments and part II the clinical study. In part I (experiments) a total of 15 samples from a previous study (MARS study) were selected that were known to be positive by conventional blood culture. To clarify this, we added the following to the results section: ‘’To test the efficacy of the MC method on bacterial DNA isolated from blood samples, a total of 15 samples were selected which had previously been found positive for nine unique pathogens’’ (line 165-167 of the revised manuscript).

In the second laboratory experiment, we used a blood sample from one healthy adult volunteer and spiked this blood with Staphylococcus haemolyticus, Escherichia coli and Proteus mirabilis. The spiked blood was split into six portions of 1ml. Three of these were pre-processed according to the Polaris protocol as described previously after which automated DNA extraction was performed on the EasyMAG machine. The other hree were directly processed with the EasyMAG machine. To clarify this, we added the following: ‘’ The spiked blood was split into six portions of 1ml. Three of these were pre-processed according to the Polaris protocol as described previously. The other three were directly processed with the EasyMAG’’ (line 184-186 of the revised manuscript).

Subsequently, we included 38 clinical patients with a suspicion of EOS as part of the clinical study. A total of 17 cases were classified as clinical EOS cases and 21 infants as uninfected controls, the used definitions were already provided in the first submission. The characteristics of these 38 participants are described in the baseline characteristics table. To clarify the number of collected peripheral and cord blood sampels we added the following sentence: ‘’In summary, we analyzed 19 peripheral blood samples and three cord blood samples from the uninfected infants. From the clinical EOS cases, we analysed 17 peripheral blood samples and one cord blood sample’’ (line 212-214 of the revised manuscript)..

We believe that, based on your suggestions, these additions clarify the two different laboratory experiments (MC testing on 15 known blood culture positive samples and the spiking experiment) and that included blood samples from 38 clinical participants in the second part of the study (clinical study).

  1. Abstract/ Introduction/ Discussion/ Conclusion: “Since MC can generate results within 4 hours following sampling compared to 36-72 hours in PBC”; “time to positivity is commonly up to 36-72 hours”; “compared to 36-72 hours in PBC” – The sentences seem to be biased in favor of MC test. Classic microbial culture on solid or liquid media takes 24 hours. If you have blood samples and there is a suspicion of systemic infection with possibility of sepsis the automated blood culture systems are used (like VITEK, BACTEC Alert), which allow to shorten a culture to a few hours. You have to give correct information about blood culture systems and time needed for the culture results.

Answer/Response of the Authors: We adjusted the sentence to time to positivity is can be up to 36-72 hour (line 40). Besides, we added information about the used blood culture system in our clinic in the method section (line 122). We agree with the reviewer that time-to-positivity may be faster than 36 hours. However, it has to be noted that it is common practise to discontinue antibiotics in case blood culture is still negative after 36-72 hours, as time-to-positivity may be prolonged in some cases. Consequently a negative blood culture result can be interpreted reliably after 36-72 hours. Using the MC, a negative result can be interpreted within 4 hours. To clarify this, we adjusted the statement made in line 319 accordingly.

  1. Materials and Methods: “classified as contamination or as clinically relevant by two independent experts (TdM, DB) – Explain the abbreviations, what kind of experts you are talking about?

Answer/Response of the Authors: This is explained in the adapted manuscript (line 150).

  1. The sentence is incomprehensible: “The sample in which MC did not detect anything was a sample with a low load of  aureus(Ct 37,6)” – Since MC is also based on PCR, you need clearly indicate the methods used (How do you know about S. aureus?).

Answer/Response of the Authors: It is explained that we selected 15 samples from a previous study (MARS study) that were known to be positive for nine different bacteria. The Ct values of these samples were determined by qPCR. To investigate the relation between the MC load and the Ct values as found by the qPCR panel, a linear regression analysis was performed between Ct values and Log2 transformed Relative Fluorescence Units of the MC. Log2 transformation was done as Ct value should also be seen as a log2 scale, as it represents measurements of the doubling cycles of PCR. Please find this explanation in line 86-90 of the revised manuscript. The Ct values and MC loads of these 15 samples are shown in table 1. 

  1. Discussion: “molecular techniques have become available for identification of bacterial DNA, such as real time PCR, 16S rRNA gene sequencing and MC” – MC is a name of company's diagnostic test (in addition disastrous and completely misleading in my opinion, but nothing can be done about it anymore), not a molecular technique (the test is based on PCR). Improve the sentence.

Answer/Response of the Authors: Molecular technique was adjusted to molecular based diagnostic tests (line 245 of the revised manuscript).

Minor comments:

  1. Whole text: Proper spelling with hyphen: Gram-positive, Gram-negative bacteria.

Answer/Response of the Authors: We have adjusted this accordingly.

  1. Table 1/ Figures / whole text: The Latin names should be written italics.

Answer/Response of the Authors: We have adjusted this accordingly.

  1. Whole text: The spaces are missing after dot in short Latin names, e.g.  coli instead of E.coli

Answer/Response of the Authors: We have adjusted this accordingly.

  1. Abstract: What do you mean writing “spiked blood samples” – combined, pooled, mixed?

Answer/Response of the Authors: In this study we performed two laboratory experiments. In the first experiment, we selected 15 culture positive samples from a previous study. In the second experiment, we used blood from a healthy adult volunteer and spiked this with bacteria. These are the spiked blood samples. The word limit in the abstract did not allow to elaborate in detail on the methods of both experiments and on the clinical study. 

  1. Abstract: “bacteria present in low concentrations” – Bacteria form a suspension, not a solution, so you can't talk about their “concentration”

Answer/Response of the Authors: concentration was adjusted to bacterial load.

  1. Whole text: “Log2” or Log(base of the logarithm)?

Answer/Response of the Authors: adjusted to Log2. 

  1. Whole text: “5ml” “1ml” etc. – lack of space before the units (it should be 5 ml, 1 ml etc.)

Answer/Response of the Authors: We have adjusted this accordingly.

  1. Materials and Methods: “PBS” -  for the first time give also the full name and the source of buffer

Answer/Response of the Authors: We have adjusted this accordingly.

  1. Materials and methods: “To test the performance of the Polaris method on small volumes of blood, we spiked 1ml aliquots of blood” – according to your later description you mixed bacterial suspension with 6 ml of blood sample and finally divided and took 1 ml for testing.

Answer/Response of the Authors: A blood sample of a healthy adult volunteer was spiked with the three bacteria, diluted with PBS and then 10 ml was added to 6 ml of blood, which was split in six portions of 1 ml. Please find the clarification in line 102-112 of the revised manuscript.

  1. Materials and methods / Discussion: “micro-organisms” – redundant dash

Answer/Response of the Authors: We have adjusted this accordingly.

  1. inbiome inBiome – unify in whole text

Answer/Response of the Authors: We have adjusted this to inbiome

  1. Whole text: Table 1 or 2, Figure 1 or 2 start from the capital letters

Answer/Response of the Authors: We have adjusted this accordingly.

  1. The legend to the Table 1 is needed: signs/descriptions (e.g. MC load) need to be explained. Clarify in what units did you present MC load”?

Answer/Response of the Authors: We have adjusted this accordingly.

Reviewer 3 Report

Comments to the author:

This is an interesting paper regarding the potential use of an innovative technique for early onset sepsis detection.

The author’s findings highlight the advantages of a quick molecular culture assay compared to the standard peripheral blood culture. 

As clearly stated among study limitations, sample size is considerably small and doesn’t allow to obtain certain conclusions. Further study with a larger and statistically powered sample size are required.

However, if these findings will be confirmed in larger studies, MC might allow an earlier diagnosis of EOS as well as a reduction of antibiotics exposure in neonates.

Several points may be added to increase the strength of this study:

  • It would be interesting if the authors may add a brief comment regarding sensitivity of this method after antibiotic administration, as previously assessed by Budding and colleagues. Was it considered in this study? If not, it could be mentioned among study limitations.

  • More objective data on sensitivity and specificity of this method should be added in the introduction paragraph.

  • A comment on the minimal volume of blood required should be added

Abstract:

  • Line 17: I would replace “overuse” into “excessive use of”

  • Line 17: I would replace “here”| with another expression

  • Line 21: I would avoid “consecutively”

  • Line 23: correct “infants” with “infant”

  • Line 24: I would better clarify this sentence

Introduction:

  • Line 35: the verb “reflect” is not appropriate in this context; you might consider instead “stem from/result from”

  • Line 45: I would add the evidence on increased risk of NEC in relation to even short period of antibiotic exposure among preterm infants or other short/medium term outcomes

  • Line 60: avoid repetition of the term “circumvent”

  • Line 70: reference(s) should be added.

  • Lines 77 to 80: the sentence is too long and therefore not clear enough. You might consider splitting the aim of the study in two different sentences.

Methods:

  • Line 87: Probably I would consider to explain the methods extensively

  • Line 105: I would probably avoid the word “On”

  • Line 112-114: I would use a chart to describe inclusion/exclusion criteria 

  • Line 114-118: I would probably add a figure with the algorithm 

  • Line 148: avoid “the”

Results:

  • Lines 153-154: I would probably change the order of the words 

  • Line 211: Avoid repetition of “furthermore”.

  • Line 213: Change th expression “suspected for EOS” 

Discussion:

  • I would change “Applicability” into “Feasibility”

  • Line 212: I would change “for” into “of”

  • Line 212: I would probably separate this sentence into 2 different ones

  • Line 219: I would change “Overuse” into “Extensive use” 

  • Line 222: I would change “Used” into “Chosen”

  • Line 223: I would change “Costly” into “Expensive”

  • Line 227: I would change “Uncoltivatable” into “Not detectable”

  • Line 241: I would change “Fastidious” into “Peculiar”

  • Line 281-283: I would probably state the characteristics of the planned study

Conclusions:

  • Line 287: I would change “To investigate” with “That investigates”

  • Line 292-294: I would suggest elaborating a Protocol related to this sentence

  • Line 297: I would substitute “at neonatology ward” with “in the neonatal population”

Author Response

Reviewer 3

Comments to the author:

This is an interesting paper regarding the potential use of an innovative technique for early onset sepsis detection.

The author’s findings highlight the advantages of a quick molecular culture assay compared to the standard peripheral blood culture. 

As clearly stated among study limitations, sample size is considerably small and doesn’t allow to obtain certain conclusions. Further study with a larger and statistically powered sample size are required.

However, if these findings will be confirmed in larger studies, MC might allow an earlier diagnosis of EOS as well as a reduction of antibiotics exposure in neonates.

Several points may be added to increase the strength of this study:

Dear Reviewer,

Thank you very much for your valuable comments on our manuscript entitled “Potential of Molecular Culture in Early-onset Neonatal Sepsis Diagnosis: a Proof of Principle Study”. Each of your insights have served to strengthen our manuscript and we have made changes to reflect them. Below we present the detailed replies to your comments and we refer to the revised manuscript.

  • It would be interesting if the authors may add a brief comment regarding sensitivity of this method after antibiotic administration, as previously assessed by Budding and colleagues. Was it considered in this study? If not, it could be mentioned among study limitations.

Answer/Response of the Authors:

Line 71-78: The sensitivity of conventional PBC for diagnosis of sepsis in neonates is being questioned. In contrast to conventional PBC, MC may detect bacteria in blood even when bacterial load is low, is not influenced by previous antibiotic exposure and is able to detect species uncultivatable by PBC.14 Therefore, it is hypothesized that MC may detect more relevant pathogens in infants with suspicion for EOS compared to a conventional PBC, especially if infants are previously exposed to antibiotics. In this study all infants were evaluated for early-onsete sepsis directly postpartum and thus not exposed to antibiotics before sampling. However, part of the mothers received intrapartum antibiotic prophylaxis, also exposing the infant to antibiotics.

Please find a comment regarding previous antibiotic exposure also in the discussion (line 272-273 of the revised manuscript).

  • More objective data on sensitivity and specificity of this method should be added in the introduction paragraph.

Answer/Response of the Authors: The analytical sensitivity is described in the IFU (which is publicly available on https://inbiome.com/wp-content/uploads/2022/12/IFU-Molecular-Culture.pdf) of the Molecular Culture assay to be 1-5 cfu with 95% confidence. Clinical sensitivity has been further demonstrated in the paper by Budding et al, 2016. In the introduction we refer to this paper. However, the sensitivity and specificity of this method have not been investigated for detection of bacteria in infants with a suspicion of early-onset sepsis, limiting to make any statements about the diagnostic accuracy for this purpose. 

  • A comment on the minimal volume of blood required should be added

Answer/Response of the Authors: Currently, there is no set minimum blood volume. It must be noted though that lower volumes yield lower sensitivity than higher volumes. We agree with the reviewer that a minimum volume of blood required should be indicated as soon as this assay is going to be used in clinical practice, however at this point this is stil too early. 

Abstract:

  • Line 17: I would replace “overuse” into “excessive use of”

Answer/Response of the Authors: We have adjusted this accordingly.

  • Line 17: I would replace “here”| with another expression

Answer/Response of the Authors: we have adjusted this to ‘in this study’

  • Line 21: I would avoid “consecutively”

Answer/Response of the Authors: A consecutive approach was used to include all neonates with EOS suspicion. However, we removed the word ‘consecutively’ as suggested.

  • Line 23: correct “infants” with “infant”

Answer/Response of the Authors: We have adjusted this accordingly.

  • Line 24: I would better clarify this sentence

Answer/Response of the Authors: The sentence was split into two sentences.

Introduction:

  • Line 35: the verb “reflect” is not appropriate in this context; you might consider instead “stem from/result from”

Answer/Response of the Authors: we have adjusted this to ‘’results from’’

  • Line 45: I would add the evidence on increased risk of NEC in relation to even short period of antibiotic exposure among preterm infants or other short/medium term outcomes

Answer/Response of the Authors: we have added NEC to the list of outcomes

  • Line 60: avoid repetition of the term “circumvent”

Answer/Response of the Authors: we have adjusted this to ‘’avoid’’

  • Line 70: reference(s) should be added.

Answer/Response of the Authors: a reference was added to this sentence.

  • Lines 77 to 80: the sentence is too long and therefore not clear enough. You might consider splitting the aim of the study in two different sentences.

Answer/Response of the Authors: The sentence was split in two.

Methods:

  • Line 87: Probably I would consider to explain the methods extensively

Answer/Response of the Authors: The Polaris method has been described previously. We added a brief explanation of this method to the method section in the revised version of the manuscript: In short, this method first degrades human cells and DNA, while pathogens remain intact. Secondly, the pathogens are pelleted via centrifugation. Then pathogens are lysed and DNA can be isolated’’ (line 90-92).

  • Line 105: I would probably avoid the word “On”

Answer/Response of the Authors: we adjusted this. The word ‘’on’’ was changed to ‘’with’’.

  • Line 112-114: I would use a chart to describe inclusion/exclusion criteria 

Answer/Response of the Authors: Infants were included when antibiotics were started for a suspicion of early-onset sepsis based on the Dutch guidelines. In this guideline, maternal risk factors and neonatal risk factors or symptoms of EOS are categorized as red flags or minor criteria. We have added two supplemental tables (supplemental Table 1 and 2) to clarify when infants are started on antibiotics and thus were eligible.

  • Line 114-118: I would probably add a figure with the algorithm 

Answer/Response of the Authors: no algorithm was used to continue or discontinue antibiotics. The decision to prolong antibiotic treatment or not was based on the clinical judgement of the treating physician. This is clarified in line 129-130. Unfortunately, there is no consensus definition for clinical early-onset sepsis. Therefore we used this definition as this is one of the most widely used definitions.

  • Line 148: avoid “the”

Answer/Response of the Authors: we adjusted this; ‘the’ was removed

Results:

  • Lines 153-154: I would probably change the order of the words 

Answer/Response of the Authors: we have adjusted this accordingly so we changed the order of the words in this sentence

  • Line 211: Avoid repetition of “furthermore”.

Answer/Response of the Authors: we have adjusted this; repetition was avoided by removing the word furthermore

  • Line 213: Change th expression “suspected for EOS” 

Answer/Response of the Authors: we have adjusted this to ‘’Furthermore, we investigated the potential of MC as diagnostic test for EOS’’

Discussion:

  • I would change “Applicability” into “Feasibility”

Answer/Response of the Authors: we have adjusted this to ‘’feasibility’’

  • Line 212: I would change “for” into “of”

Answer/Response of the Authors: The sentence was adjusted

  • Line 212: I would probably separate this sentence into 2 different ones

Answer/Response of the Authors: The sentecene was adjusted;

  • Line 219: I would change “Overuse” into “Extensive use” 

Answer/Response of the Authors: we have adjusted this;

  • Line 222: I would change “Used” into “Chosen”

Answer/Response of the Authors: we have adjusted this accordingly;

  • Line 223: I would change “Costly” into “Expensive”

Answer/Response of the Authors: we have adjusted this accordingly;

  • Line 227: I would change “Uncoltivatable” into “Not detectable”

Answer/Response of the Authors: we have adjusted this accordingly;

  • Line 241: I would change “Fastidious” into “Peculiar”

Answer/Response of the Authors: we have adjusted this accordingly;

  • Line 281-283: I would probably state the characteristics of the planned study

Answer/Response of the Authors: we have adjusted this accordingly (line 308 of the revised manuscript);

Conclusions:

  • Line 287: I would change “To investigate” with “That investigates”

Answer/Response of the Authors: we have adjusted this accordingly;

  • Line 292-294: I would suggest elaborating a Protocol related to this sentence

Answer/Response of the Authors: ‘’Since MC can generate results within 4 hours following sampling, compared to 36-72 hours in PBC, MC may guide clinicians faster to discontinue antibiotic therapy in case of a negative MC test and reassuring clinical condition of the infant’’ was adjusted to:

‘’Since MC can generate results within 4 hours following sampling, compared to 36-72 hours in PBC, MC may replace PBC in the current guideline. This may guide clinicians faster to discontinue antibiotic therapy in case of a negative MC test and reassuring clinical condition of the infant’’. Besides, in the discussion, we briefly elaborate on the details of this potential new protocol.  However, as this is a proof of principle study, our findings should first be validated in a large cohort with statistically powered sample size that MC can replace the conventional blood culture in current guidelines. We therefore believe it is too yet too early to go in much detail on a potential new protocol in this manuscript including MC instead of a conventional blood culture. We fully agree this new protocol is of great interest following the step of validation

  • Line 297: I would substitute “at neonatology ward” with “in the neonatal population”

Answer/Response of the Authors: we have adjusted this accordingly;

Round 2

Reviewer 2 Report

Almost all of my comments have been thoughtful, explained and included in revised text. I am not entirely convinced by your explanation of the timing of the results from blood culture. Moreover, I still have one comment concerning the revised “Conclusion”, in which you suggest that “MC may replace PBC in the current guideline”. I am convinced that a complete replacement of PBC will not be possible, if only because of the need for blood culture in case of a positive result followed by antibiogram. Therefore, I recommend strongly softening and/or clarifying this statement. Thus, in my opinion the manuscript can be published in Microorganisms after minor revision.

Author Response

Almost all of my comments have been thoughtful, explained and included in revised text. I am not entirely convinced by your explanation of the timing of the results from blood culture. Moreover, I still have one comment concerning the revised “Conclusion”, in which you suggest that “MC may replace PBC in the current guideline”. I am convinced that a complete replacement of PBC will not be possible, if only because of the need for blood culture in case of a positive result followed by antibiogram. Therefore, I recommend strongly softening and/or clarifying this statement. Thus, in my opinion the manuscript can be published in Microorganisms after minor revision.

Dear reviewer,

Thank you very much for your time to review the revised version of our manuscript ‘’Potential of Molecular Culture in Early-onset Neonatal Sepsis Diagnosis: a Proof of Principle Study’’. 

We agree with the reviewer that the investigated molecular culture technique does not provide antibiotic resistance patterns. Therefore we added the following to the conclusion: ‘’As long as there are no opportunities to establish antibiotic resistance patterns by novel molecular techniques, blood cultures will be necessary for this purpose’’ (line 277-280).

Besides, we acknowledge this in the discussion by adding: ‘’However, it has to be acknowledged that Molecular Culture does not provide antibiotic resistance patterns. Therefore, conventional blood cultures will be necessary for this purpose as long as there are no opportunities to establish antibiotic resistance patterns by novel molecular techniques’’ (line 311-316).

Furthermore, in the revised manuscript we tried to stress that conventional blood cultures may be positive rapidly, but negative results can be interpreted reliably after 36-72 hours and therefore infants with negative blood cultures are currently exposed for 36-72 hours. When using Molecular Culture, negative culture results can be interpreted within 4 hours.
